# Slide-SAM: Medical SAM Meets Sliding Window

**Quan Quan**[*1,2]                                        QUAN.QUAN@MIRACLE.ICT.AC.CN
[1] *Institute of Computing Technology, Chinese Academy of Sciences*
[2] *University of Chinese Academy of Sciences*

**Fenghe Tang**[*3,4]                                         FHTAN9@MAIL.USTC.EDU.CN
**Zikang Xu**[3,4]                                          ZIKANGXU@MAIL.USTC.EDU.CN
**Heqin Zhu**[3,4]                                          ZHUHEQIN@MAIL.USTC.EDU.CN
**S.Kevin Zhou**[1,2,3,4]                                    SKEVINZHOU@USTC.EDU.CN
[3] *School of Biomedical Engineering, Division of Life Sciences and Medicine, University of Science and Technology of China*
[4] *Suzhou Institute for Advanced Research, University of Science and Technology of China*

**Editors:** Accepted for publication at MIDL 2024

## Abstract

The Segment Anything Model (SAM) has achieved a notable success in two-dimensional image segmentation in natural images. However, the substantial gap between medical and natural images hinders its direct application to medical image segmentation tasks. Particularly in 3D medical images, SAM struggles to learn contextual relationships between slices, limiting its practical applicability. Moreover, applying 2D SAM to 3D images requires prompting the entire volume, which is time- and label-consuming. To address these problems, we propose Slide-SAM, which treats a stack of three adjacent slices as a prediction window. It firstly takes three slices from a 3D volume and point or bounding box prompts on the central slice as inputs to predict segmentation masks for all three slices. Subsequently, the masks of the top and bottom slices are then used to generate new prompts for adjacent slices. Finally, step-wise prediction can be achieved by sliding the prediction window forward or backward through the entire volume. Our model is trained on multiple public and private medical datasets and demonstrates its effectiveness through extensive 3D segmetnation experiments, with the help of minimal prompts. Code is available at https://github.com/Curli-quan/Slide-SAM.

**Keywords:** Segmentation, Interactive segmentation.

## 1. Introduction

Nowadays, 3D medical image segmentation plays a crucial role in medical image analysis for clinical analysis and diagnosis. However, annotating 3D medical images requires a significant amount of human labor and time resources. Recently, the Segment Anything Model (SAM) (Kirillov et al., 2023) demonstrates impressive zero-shot segmentation capabilities in large-scale computer vision tasks (Mazurowski et al., 2023; Huang et al., 2023b; He et al., 2023). In the field of medical imaging, SAM introduces new possibilities for accelerating data annotation by using non-fixed points, bounding boxes, and rough mask prompts to define segmentation region categories. However, the huge distributional gap between natural images and medical images makes SAM inapplicable directly to medical images (Huang

---

[*] Contributed equally

et al., 2023a; Cheng et al., 2023). One straightforward solution to bridge the gap is finetuning SAM using medical images (Cheng et al., 2023; Ma and Wang, 2023; Wu et al., 2023). Another feasible approach is to adjust the numeric range of medical images, making them visually more akin to natural images, which greatly improves the segmentation ability of SAM in the medical domain. Given these straightforward solutions, we believe that this issue may not be the central challenge in medical segmentation tasks.

The fundamental challenge that SAM faces with medical images, in our view, lies in its inability to efficiently segment 3D images. Most recent variants can only employ a slice-by-slice approach for processing volumetric images (Cheng et al., 2023; Ma and Wang, 2023). However, such methods require substantial manual assistance and overlook the contextual information between slices, resulting in evident discontinuities in the segmentation results on each layer. Some other methods utilize adapters to introduce information between slices (Bui et al., 2023; Lei et al., 2023; Wu et al., 2023) whereas they still require a substantial cost in terms of prompt annotations. Additionally, there exists another category of methods that directly extend SAM into a 3D model (Gong et al., 2023). Nevertheless, this method relinquishes the assistance provided by SAM's pre-trained weights due to the difference between tasks, necessitating a more resource-intensive model training process. Indeed, a core question prominently arises: *How can SAM be enabled to predict 3D data effectively with only one prompt while fully harnessing its pre-trained weights?* To address it, we propose a network called Slide-SAM. Slide-SAM only requires a prompt from the central slice to simultaneously infer multiple adjacent slices, and the resulting predictions can be used to generate prompts for the next group of adjacent slices. This is achieved through a sliding window approach, ultimately enabling one prompt to segment an entire volume. Furthermore, Slide-SAM's architecture and task are similar to the original SAM (from an RGB image to 3 grayscale images), making it easier to leverage the pre-trained weights of SAM. Regarding data, we utilize both 3D ground truth labels and 2D pseudo-labels generated by SAM and we introduce a hybrid loss function that controls which slices to calculate the loss on, allowing the incorporation of single-slice labels and multi-slice labels. Extensive experiments prove that our Slide-SAM can gain superior inference performance on 3D images with minimal prompt cost.

## 2. Method

### 2.1. Structure of SAM

SAM consists of three main components: image encoder, prompt encoder and light-weight mask decoder. The image encoder is based on the Vision Transformer (ViT) (Dosovitskiy et al., 2020) pretrained by MAE (He et al., 2022) to extract representations. Prompt encoders can handle both sparse (points, boxes, text) and dense (masks) prompts. In this paper, we mainly focus on sparse encoders, which represent points and boxes as positional encodings that are then summed with the learned embeddings for each prompt type. The mask decoder is a Transformer decoder block modified to include dynamic mask prediction headers. SAM uses bidirectional cross-attention in each block, one for prompt-to-image embeddings and the other for image-to-prompt embeddings, to learn the interaction between prompt and image embeddings. After fusing two embeddings, SAM upsamples the image

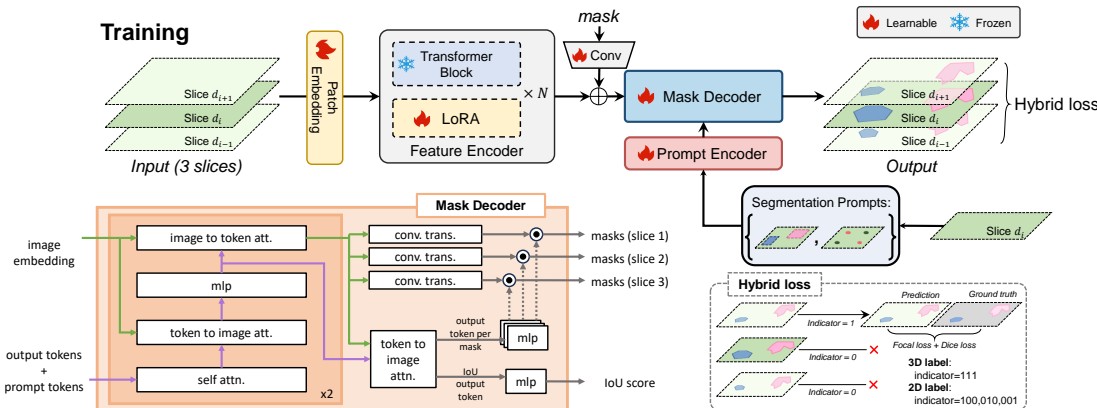

Figure 1: The training pipeline of Slide-SAM. First, Three adjacent slices are used as input and fed into the backbone network. Then, the Prompt encoder is employed to encode points or boxes. The Mask decoder receives the generated features from the previous step and generates masks for each slice using different heads. The hybrid loss is only computed for layers with labels.

embedding, and then a multilayer perceptron maps the output labels to a dynamic linear classifier that predicts the target mask given the image.

## 2.2. Structure of Slide-SAM

**Feature Encoder:** Concerning the feature encoder, which is a transformer encoder, we incorporate LoRA (Hu et al., 2021), enabling SAM to update only a small subset of parameters during medical image training. The ViT-B (used in SAM-B) serves as the backbone and remains frozen throughout training, while the embedding layer is configured to be trainable.
**Mask Encoder:** Given an image with 3 slices $X \in \mathbb{R}^{H \times W \times 3}$, the mask decoder efficiently maps the image embedding $F_{im} \in \mathbb{R}^{c \times h \times w}$, prompt embeddings $P \in \mathbb{R}^{5 \times c}$, and an output token into feature maps $F_o \in \mathbb{R}^{c \times h \times w}$ representing the images, and three prediction heads $H$ representing the prompted-based tasks. In order to segment multiple slices at the same time, the feature maps are expanded from one to three, but followed with the same heads.

Specifically, for the feature map $F_o$, Slide-SAM has three different MLP blocks to convert the feature map $F_o$ into three feature maps $F_1 \in \mathbb{R}^{c \times h \times w}$, $F_2 \in \mathbb{R}^{c \times h \times w}$, and $F_3 \in \mathbb{R}^{c \times h \times w}$. Each feature map represents a slice. The heads $H$ are divided into three heads $H_{s1} \in \mathbb{R}^c$, $H_{s2} \in \mathbb{R}^c$, and $H_{s3} \in \mathbb{R}^c$ for segmenting different semantic regions, and $H_u \in \mathbb{R}^c$ for IoU prediction. Similar to SAM, we can obtain mask predictions $M \in \mathbb{R}^{H \times W \times 3 \times 3}$ and IoU predictions $U \in \mathbb{R}^3$:

$$
\begin{aligned}
M_{ij} &= F_i \odot H_{sj}, \quad j = \{1, 2, 3\} \\
U_j &= MLP(H_u),
\end{aligned}
\tag{1}
$$

where $\odot$ is point-wise multiplication. Additionally, to fully leverage the prior knowledge of SAM weights, we load all weights from the Transformer decoder and the weights of the heads. We duplicate the weights of the MLP block receiving the feature map $F_o$ from one to three to load into the three branches of Slide-SAM.

**Prompt Encoder:** We consider two sets of prompts: sparse (points, boxes) and dense (masks). The distinction between our method and SAM lies in the fact that our input images consist of three slices, and we opt to select the middle slice as a reference to provide prompts. Regarding the mask prompt, we extend the input channel count of the convolutional block associated with the mask prompt to three, in order to facilitate the input of masks from three layers of slices. Points and boxes are encoded by positional encodings summed with learned embeddings for each prompt type. Mask prompts are encoded and then summed element-wise with the image embedding.

## 2.3. Training strategy

**Hybrid loss:** Given the prediction $M \in \mathbb{R}^{H \times W \times 3 \times 3}$ and the ground-truth $\hat{M} \in \mathbb{R}^{H \times W \times 3}$, Slide-SAM adopts cross-entropy and Dice loss to supervise the fine-tuning process. The loss function can be described as follows:

$$
\begin{aligned}
\mathcal{L}_{seg}^{j}(M_j, \hat{M}) &= \lambda_1 \mathcal{L}_{ce}(M_j, \hat{M}) + \lambda_2 \mathcal{L}_{dice}(M_j, \hat{M}), \\
\mathcal{L}_{iou}^{j}(U_j, M_j, \hat{M}) &= \mathcal{L}_{mse}(U_j, IoU(M_j, \hat{M}))
\end{aligned}
\tag{2}
$$

$$
\mathcal{L} = \mathcal{L}_{seg}^{k} + \mathcal{L}_{iou}^{k}, \quad k = \arg\min_{j} \mathcal{L}^{j},
\tag{3}
$$

where $\mathcal{L}_{ce}$ and $\mathcal{L}_{dice}$ represent cross-entropy loss and dice loss, respectively. $M \in \mathbb{R}^{H \times W \times 3}$ and $\hat{M} \in \mathbb{R}^{H \times W \times 3}$ represent the prediction and the ground truth, respectively. $\lambda_1$ and $\lambda_2$ represent loss weights, which are used to balance the impact between these two loss terms. $\lambda_1$ and $\lambda_2$ are set to 20 and 1 in practice, respectively.

Next, to facilitate concurrent training on 2D and 3D data, we introduce an indicator $\mathcal{I} \in \{0, 1\}^3$ to guide the layers for which loss computation is required. As a result, Eq (3) can be transformed as follows:

$$
\hat{\mathcal{L}}_{seg}^{j} = \mathcal{I}\mathcal{L}_{seg}(M_j, \hat{M}),
\tag{4}
$$

For instance, when using 3D labels, all three slices possess masks, resulting in each value of the indicator being 1. Conversely, when using 2D labels, only one slice contains a mask, leading to the indicator values being set to 1 for the slice with the mask and 0 for the others. In the case of IoU prediction loss, since it is not feasible to accurately predict all masks when using 2D labels, the exact IoU values remain unknown. Therefore, in such situations, we set the IoU prediction loss to be 0.

## 2.4. Inference

Our inference process, as shown in Figure 2, begins by selecting a specific layer as the starting point. We provide prompts for points or bounding boxes on this layer. Subsequently, we input this layer and its adjacent slices (3 slices in total) into the model to obtain segmentation masks. The masks at both ends are then denoised by morphological opening and used for computing bounding boxes for the next 3 consecutive slices. We iterate through the inference process in a **sliding window** fashion, extending towards both ends, until one of the masks at ends is empty.

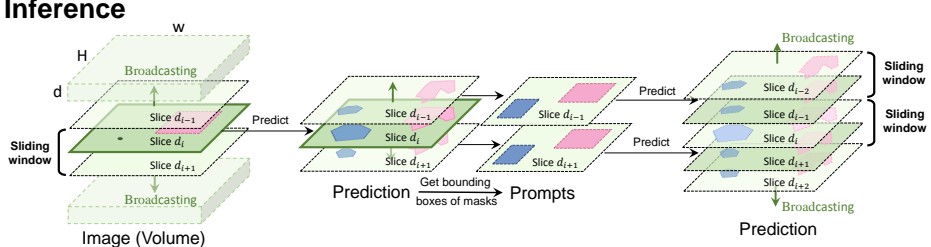

Figure 2: The inference process of Slide-SAM.

| Method | CHAOS | | | | | BTCV | | | | |
|---|---|---|---|---|---|---|---|---|---|---|
| | Liver | Kid(R) | Kid(L) | Spleen | Avg. | Liver | Stomach | Kid(L) | Spleen | Avg. |
| nnUNet (Isensee et al., 2021) | 87.95 | 93.91 | 93.67 | 87.78 | 90.82 | 96.17 | 76.79 | 94.81 | 93.84 | 90.40 |
| SSL-ALP (Ouyang et al., 2020) (1 volume) | 63.82 | 56.52 | 63.68 | 73.40 | 64.35 | 69.40 | 34.35 | 44.89 | 65.00 | 53.40 |
| SAM (N points) | 29.79 | 36.93 | 63.77 | 45.55 | 44.01 | 39.60 | 31.74 | 68.57 | 23.46 | 40.84 |
| SAM-Med2d (N points) | 58.48 | 89.16 | 87.53 | 72.51 | 76.92 | 84.89 | 79.02 | 92.97 | 92.03 | 87.22 |
| SAMMed-3D (N points) | 82.80 | 80.25 | 78.49 | 79.37 | 80.22 | 64.31 | 54.73 | 85.98 | 75.38 | 70.10 |
| Ours (1 point) | 88.39 | 91.86 | 90.74 | 90.53 | 90.38 | 92.38 | 89.03 | 92.35 | 75.62 | 87.34 |
| SAM (N boxes) | 78.48 | 93.81 | 92.40 | 91.84 | 89.13 | 72.18 | 83.98 | 93.67 | 88.51 | 84.58 |
| SAM-Med2d (N boxes) | 90.09 | **94.39** | **94.03** | 92.73 | **92.81** | 93.58 | 82.88 | 93.49 | **94.07** | 91.00 |
| Ours (1 box) | 88.42 | 91.89 | 90.98 | 90.76 | 90.51 | 95.44 | 74.31 | 92.42 | 91.16 | 88.33 |
| Ours (5 boxes) | 91.70 | 92.18 | 91.37 | 92.03 | 91.82 | 95.62 | 90.88 | 93.49 | 91.88 | 92.96 |
| Ours (N boxes) | **92.94** | 92.40 | 91.60 | **92.75** | 92.42 | **95.75** | **93.24** | **93.52** | 93.44 | **93.98** |

Table 1: Evaluation on CHAOS and BTCV testsets (**Dice (%)**).

## 3. Experiment

### 3.1. Dataset preparation

**Training data:** The training data we used is divided into two parts. (1) Annotated datasets. The public datasets include AbdomenCT-1K (Ma et al., 2021), TotalSegmentor (Wasserthal et al., 2023), CTPelvic1K (Liu et al., 2021), WORD (Luo et al., 2021), etc. and some private data; (2) Pseudo-labels generated by SAM. We use SAM to generate labels for unlabeled or partially labeled data. These labels are typically in 2D format and need to be used in conjunction with the mixed loss function we propose.

**Evaluation settings:** We choose ISBI 2019 Combined Healthy Abdominal Organ Segmentation Challenge (CHAOS) (Kavur et al., 2021), MICCAI 2015 Multi-Atlas Abdominal Labeling challenge (BTCV) (Landman et al., 2015) and WORD (Luo et al., 2021) as the validation dataset. CHAOS and BTCV are split into training and test sets in a 4:1 ratio. 20 cases are taken from the WORD as testset. MSD Pancreas and MSD Colon (Antonelli et al., 2022) are split into training, validation and test sets in a 7:1:2 ration, following (Gong et al., 2023). We choose **Dice** as the evaluation metric.

**Preprocessing:** We initially apply value clipping to confine the range of CT data within [-200, 400] and MRI data within [0, 600]. Subsequently, we standardize the intensity values of each volume to the range [0, 255]. We proceed to extract all slice images along the x, y, and z axes, along with their corresponding masks. These slices are then organized and saved in groups of three adjacent slices. During the extraction process, we discard groups for which the percentage of the central slice's mask area is less than 0.14%.

**Settings:** To quantitively assess the performance of our model, we conduct comparative experiments involving fully-supervised networks, including nnUNet (Isensee et al., 2021), a one-shot network SSL-ALp (Ouyang et al., 2020), as well as SAM (Kirillov et al., 2023),

| Type | Methods | Pancreas Tumor | | | Colon Tumor | | |
|---|---|---|---|---|---|---|---|
| Supervised models | nnUNet (Isensee et al., 2021)[1] | 41.65 | | | 43.91 | | |
| | nnFormer (Zhou et al., 2023)[1] | 36.53 | | | 24.28 | | |
| | Swin-UNETR (Hatamizadeh et al., 2021)[1] | 40.57 | | | 35.21 | | |
| SAM variants | Num of Prompts | 1 | 3 | 10 | 1 | 3 | 10 |
| | SAM-B (Kirillov et al., 2023)[1] | 24.01 | 29.80 | 30.55 | 28.83 | 35.26 | 39.14 |
| | SAM3d-Adapter (Gong et al., 2023)[1] | 54.09 | 54.92 | 57.47 | 48.35 | 49.43 | 49.99 |
| | Ours | **60.28** | **70.01** | **80.09** | **61.89** | **69.80** | **71.55** |

[1] Copied from (Gong et al., 2023).

Table 2: Comparison with classical medical image segmentation methods on MSD Pancreas and MSD Colon datasets (**Dice (%)**).

| Method | Prompt | Liver | Spleen | Kid(L) | Kid(R) | Stomach | Gallbladder | Esophagus | Pancreas |
|---|---|---|---|---|---|---|---|---|---|
| SAM-B | ∼40 boxes | 74.98 | 89.74 | **93.57** | **93.56** | 78.86 | 82.21 | **72.09** | 62.31 |
| SAM-Med2d | ∼40 boxes | 93.51 | 92.00 | 91.66 | 92.01 | 87.48 | 70.68 | 52.07 | 64.81 |
| SAMMed-3D | 1 point | 90.38 | 82.83 | 84.10 | 87.24 | 55.28 | 60.56 | 28.02 | 43.49 |
| Ours | 1 box | 94.50 | 87.36 | 92.86 | 92.45 | 76.40 | 73.75 | 37.42 | 61.65 |
| Ours | 5 boxes | **95.55** | 92.03 | 92.32 | 92.73 | 91.35 | **83.01** | 48.97 | 65.54 |
| Ours (ViT-H) | 1 box | 94.36 | 90.01 | 93.04 | 91.39 | 76.46 | 69.03 | 60.75 | 71.84 |
| Ours (ViT-H) | 5 boxes | 95.25 | **92.46** | 93.40 | 92.60 | **92.08** | 78.57 | 69.52 | **82.38** |
| Method | Duodenum | Colon | Intestine | Adrenal | Rectum | Bladder | Femur(L) | Femur(R) | **Avg.** |
| SAM-B | 59.03 | 34.99 | 54.98 | 17.21 | 84.05 | 86.23 | 89.53 | 89.08 | 72.65 |
| SAM-Med2d | 58.10 | 54.54 | 73.32 | 15.45 | 85.37 | 91.71 | 89.40 | 89.24 | 75.14 |
| SAMMed-3D | 31.41 | 24.45 | 30.47 | 0.86 | 60.38 | 86.84 | 83.89 | 73.65 | 57.74 |
| Ours | 44.75 | 31.69 | 50.92 | 24.80 | 73.92 | 90.85 | 89.12 | 90.33 | 69.54 |
| Ours | 62.89 | 64.33 | 76.07 | 32.54 | 76.94 | **92.88** | 92.04 | 92.60 | 78.23 |
| Ours (ViT-H) | 50.00 | 50.15 | 56.78 | 33.98 | 77.86 | 84.11 | 93.47 | 93.72 | 74.18 |
| Ours (ViT-H) | **76.70** | **79.43** | **81.77** | **49.97** | **86.68** | 91.76 | **93.89** | **93.99** | 84.38 |

Table 3: Evaluation on WORD testset with *box prompts* (**Dice (%)**). The **best** and second best are highlighted

SAM-Med2d (Cheng et al., 2023), SAM3d-Adapter (Gong et al., 2023). The fully supervised networks employ all training data and corresponding labels. SAM and SAM-Med2d are initialized with public weights. The one-shot segmentation model utilizes training data but abstains from using labels, and requires only the labels of one volume to make predictions.

**Implementation details:** We use AdamW as the optimizer, and set the training rate to 0.0002. $\beta_1$, $\beta_2$ and weight decay settings are 0.9, 0.999 and 0.1 respectively. We end training at 20 epochs. Our models are trained on 4 Nvidia RTX GPUs with 24G GPU memory.

### 3.2. Main results

In Table 1, we find that our performances are much better than SAM and SAM-Med2d when using point prompts, and compared to them using prompts on each slice, we only use one point prompt. When using 1 box prompt, we achieve similar performance to SAM-Med2d on multiple anatomies. In addition, when we also use box prompts on each slice like SAM, we can achieve the best performance on multiple anatomies. While initial performance gains are significant, further improvements plateau as the number of prompts increases. This is because our method automatically generates reliable prompts for other slices, minimizing the marginal benefit of additional prompts.

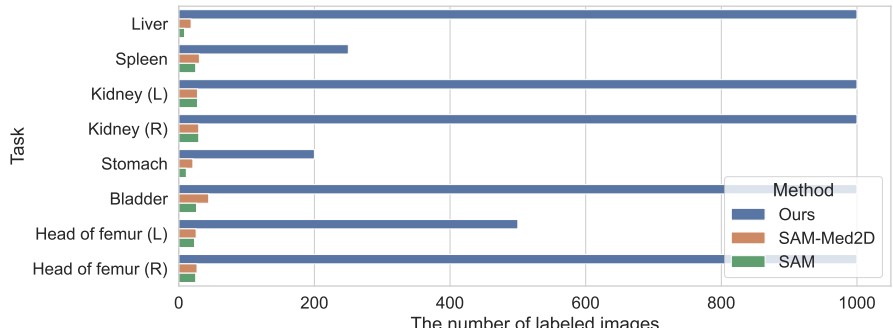

Figure 3: Labeling efficiency: The number of images that can be annotated using 1000 prompts for WORD testset.

For tumor detection, we compare our method with SAM3d-Adapter (Gong et al., 2023) and some popular segmentation methods, nnU-Net (Isensee et al., 2021), nnFormer (Zhou et al., 2023), Swin-UNETR (Hatamizadeh et al., 2021). We finetune Slide-SAM with tumor datasets based on weights well-trained on fore-mentioned large-scale medical datasets. As shown in Table 2, our performance Our model significantly outperforms existing supervised methods and SAM models, while requiring only a small number of prompts. This can greatly improve annotation efficiency.

For the WORD testset, as illustrated in Table 3, we achieve competitive results with only 5 prompts per anatomical structure, in stark contrast to SAM and SAM-Med2d, which necessitate approximately 40 prompts on average for each structure. Moreover, we leverage a larger backbone network (ViT-H) and incorporate additional CT data to train Slide-SAM, observing a more robust performance.

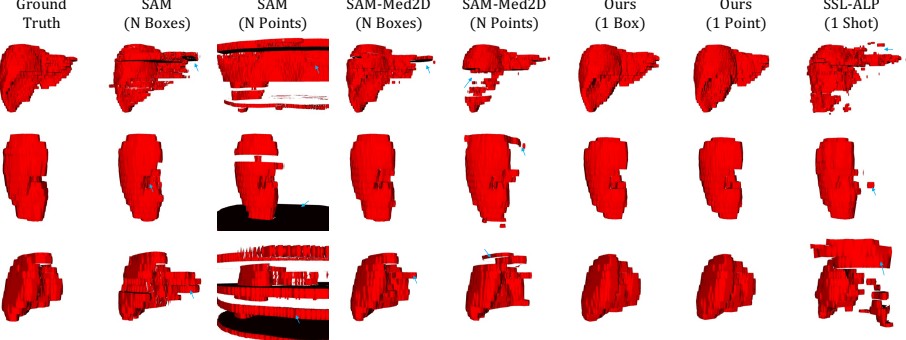

Figure 4: Visual comparison on the CHAOS dataset.

### 3.3. Other analysis

**Analysis of prompt efficiency:** Assuming we set the condition for successful annotation as having a Dice coefficient greater than 0.9 between predictions and ground-truth labels, we counted the number of images successfully annotated by our method and other methods

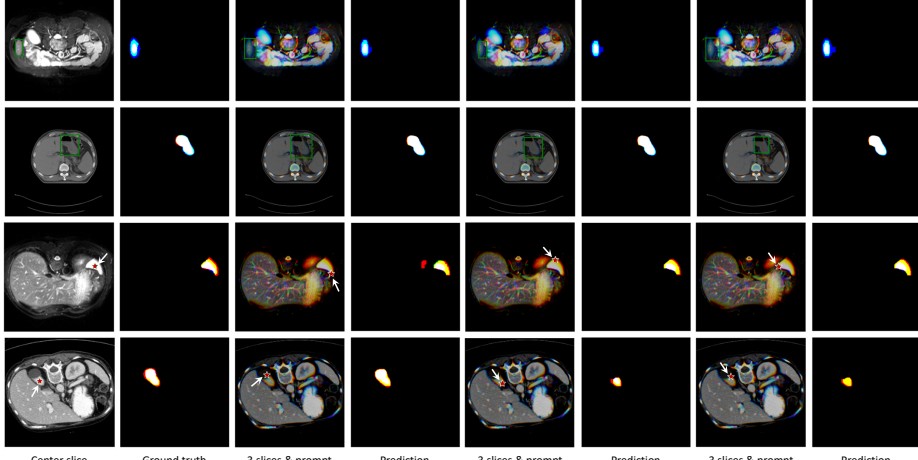

|  Center slice | Ground truth | 3 slices & prompt | Prediction | 3 slices & prompt | Prediction | 3 slices & prompt | Prediction |

Figure 5: Predictions of BTCV testset with different *noisy* prompts. We display 3 slices and their masks in RGB format.

(SAM, SAM-Med2D) when using the same 1000 prompts. As demonstrated in Figure 3, the annotation efficiency of our method is significantly higher than that of other methods.

**Visual comparison:** As depicted in Figure 4, the segmentation results from the original SAM exhibit noticeable discontinuities between upper and lower layers, leading to incoherence between adjacent layers. Additionally, when using point prompts, a significant number of segmentation errors are prevalent, causing considerable challenges for experts. In contrast, our method produces remarkably smooth results, whether utilizing point prompts or box prompts as initial prompts. It provides excellent 3D segmentation results and facilitates subsequent annotation optimization, making it a more user-friendly option for experts.

**Noisy prompts:** As shown in Figure 5, we attempt to simulate a realistic annotation environment by using points or bounding box prompts with noise. We find that our method exhibits a certain level of robustness to noise. For box prompts, stable prediction results are obtained regardless of translation or scaling. The stability of point prompts is relatively lower. Poor predictions may occur with points at edges.

## 4. Conclusion

One main challenge AM encounters when applied to medical images mainly stems from its limitations in effectively segmenting 3D image data. To address this, we introduce Slide-SAM, which leverages pretrained weights and facilitates multi-slice inference through a sliding window technique. Incorporating our data enrichment strategies and a hybrid loss function that encompasses both 3D labels and 2D pseudo-labels, our method enhances the training process and results in performance advancements. Extensive experiments prove that our Slide-SAM can gain superior inference performance on 3D images with a minimal prompt cost.

## 5. Acknowledgements

Supported by Natural Science Foundation of China under Grant 62271465, Suzhou Basic Research Program under Grant SYG202338, and Open Fund Project of Guangdong Academy of Medical Sciences, China (No. YKY-KF202206).

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

# Appendix A.  Usage for clinicians

To assist clinicians in using our tool more easily, we provide a detailed overview of the workflow for our method. We plan to integrate our tool into existing annotation software. Assuming the annotation tool with our method is presented as follows, the process begins by dragging the data needing annotation into the tool. Then, a suitable slice is selected, and an enclosing box is drawn around the desired area for annotation. For further refinement of the annotation results, additional point prompts can be added, or a more suitable bounding box can be drawn on slices where other segmentation results are unsatisfactory. We recommend ensuring that the box prompt wraps around the target as much as possible to avoid missing cases. As our method has certain performance limitations, if the desired results are not achieved, users can modify the labels manually using the annotation tool.

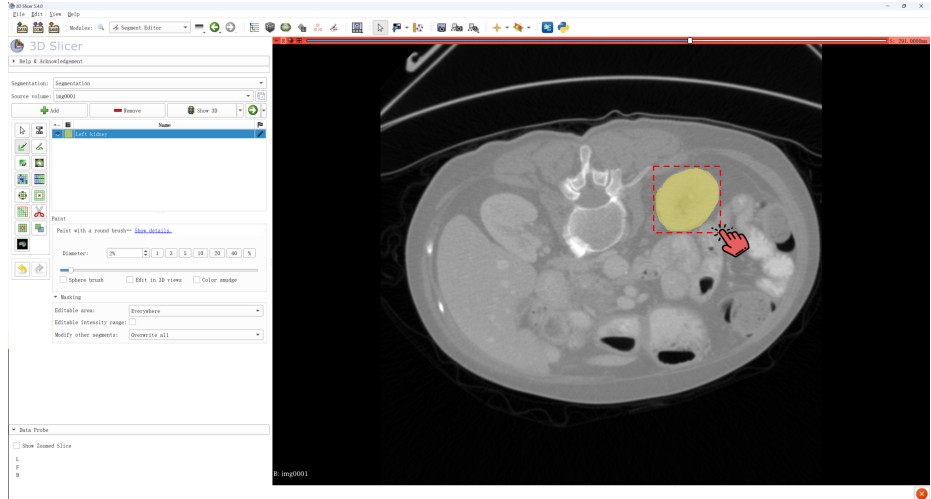

Figure 6: Example of labeling an anatomy.  Drawing a bounding box of the target on a slice.

# Appendix B.  Dataset preparation

The training data we used is divided into two parts.

- Annotated public datasets and private datasets. The public datasets include AbdomenCT-1K (Ma et al., 2021), Total Segmentor (Wasserthal et al., 2023), CTPelvic1K (Liu et al., 2021), WORD (Luo et al., 2021), etc. and some private data.

- Pseudo-labels generated by SAM. As shown in Figure 7, we use SAM to generate labels for unlabeled or partially labeled data. These labels are typically in 2D format and need to be used in conjunction with the mixed loss function we propose.

Annotated datasets: We collect multiple datasets including over 4000 CT and MRI volumes and over 30,000 3D masks. We segment all 3D volumes and labels into sets of three consecutive slices, and resized them to (1024, 1024). The images are stored in JPG format with compression, while the labels are stored as sparse matrices.

Pseudo-labels: Since some datasets have only partial annotations, we employ a straightforward method to generate a large number of pseudo-labels and apply them after training. Moreover, we observe that these data indeed result in a significant performance improvement. The generation of pseudo-labels is as follows: We find that by adjusting the window width of CT or MRI images (i.e., adopting different truncation methods, such as constraining data within the range of [-200, 400] for CT), SAM can produce different results for the same data. We believe that this adjustment can make certain regions, originally with small color differences, more distinguishable, allowing SAM to segment these areas. Therefore, we used multiple truncation thresholds, $[\mu \pm 3 * \delta]$, $[\mu \pm 2 * \delta]$, $[\mu \pm \delta]$, and $[\mu \pm 0.5 * \delta]$, where $\mu$ and $\delta$ refer to the average and standard variance, respectively. In addition, we used superpixels to generate point/box prompts for SAM. We would exclude superpixels with an average value below a certain threshold, as we consider them potentially representing background. Figure 7 illustrates the pseudo-labels we generate.

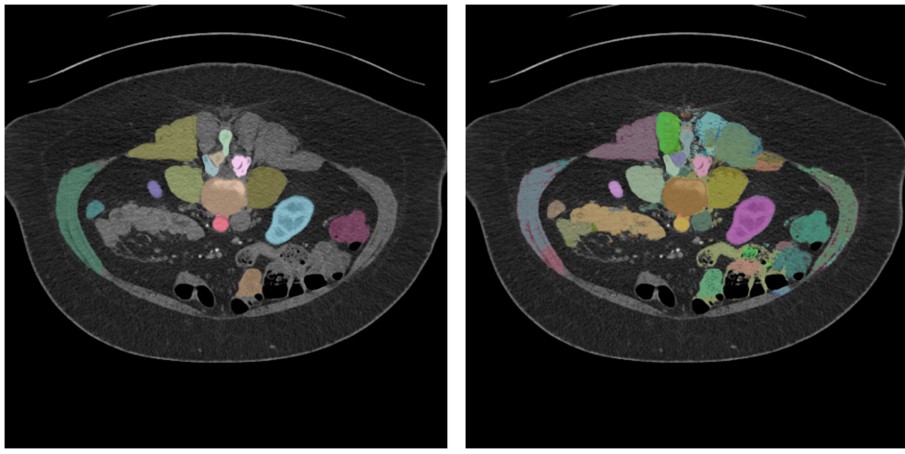

Figure 7: Pseudo labels on an image from AbdomenCT-1K dataset. (left) GT + Pseudo labels with different value ranges; (right) + superpixel-prompted pseudo labels.

## Appendix C. Implementation details

**Prompt genetation:** Following SAM and other interactive segmentation models, we simulate an interactive segmentation setup during training. First, with equal probability, either a foreground point or bounding box is selected randomly for the target mask. Points are sampled uniformly from the ground truth mask. Boxes are taken as the ground truth mask's bounding box, with random noise added in each coordinate with a standard deviation equal to 10% of the box sidelength, to a maximum of 20 pixels. This noise profile is a reasonable compromise between applications like instance segmentation, which produces a tight box around the target object, and interactive segmentation, where a user may draw a loose box.

### C.1. Inference

**2D post-processing:**

- (a) Filter out areas with IoU predictions less than 0.4;

- (b) Filter out areas with stability scores less than 0.6. The stability score is calculated as follows: given a certain stable interval such as [-0.1, 0.1], add the corresponding offset to the original logits and check the changes in the prediction area. Stability score = smallest prediction area/largest prediction area. Areas with a stability score less than a certain value will be filtered.

- (c) Calculate circumscribed matrices for each mask, and use non-maximum suppression (NMS) to remove overlapping masks using all matrices and their corresponding prediction confidence values as input.

**3D post-processing (sliding window):** First, we predict the results for the central slice. Then, the iterative inference process splits into two directions: forward and backward. For instance, in the forward direction,

- (a) We utilize the masks on the first slice of each predicted slice result. We apply morphological opening to denoise each mask and compute bounding boxes. These bounding boxes serve as prompts for another round of segmentation using the model. In this segmentation step, the central slice is the one associated with the prompt.

- (b) For areas on this slice that lack coverage from existing masks, we evenly sample points as prompts for segmentation, following the same segmentation procedure as described earlier.

- (c) All obtained masks are then subjected to the filtering method described earlier. Subsequently, the process continues with shifting and predicting masks in the specified direction. The operations in the backward direction are similar to those outlined above.

**Parallel strategy:**

We adopt a parallel strategy during inference, that is, divide all adjacent slices into multiple batches, and increase the batch size as large as possible to ensure that as much GPU memory is utilized as possible. Inference is performed on Nvidia Titan RTX 24G and the batchsize is set to 4. For Slide-SAM, the running speed is 3.25 sec/volume, and the GPU memory usage is 14G. For Slide-SAM-H, the running speed is about 10 sec/volume, and the GPU memory usage is 17G. In future, we will also try more mobile-friendly technologies to light-weight and accelerate model inference, such as using EfficientViT as backbone or SAMI pretrain strategy in EfficientSAM, using model compression and distillation technology or other operator acceleration technologies, etc.

## Appendix D. Other Analysis

### D.1. Noisy Prompts

We test more noisy prompts as shown in Figure 8. Here, we mainly test box prompts because the results obtained from box prompts are more robust. Box prompts are our primary annotation method. We find that areas not covered by box prompts are easily

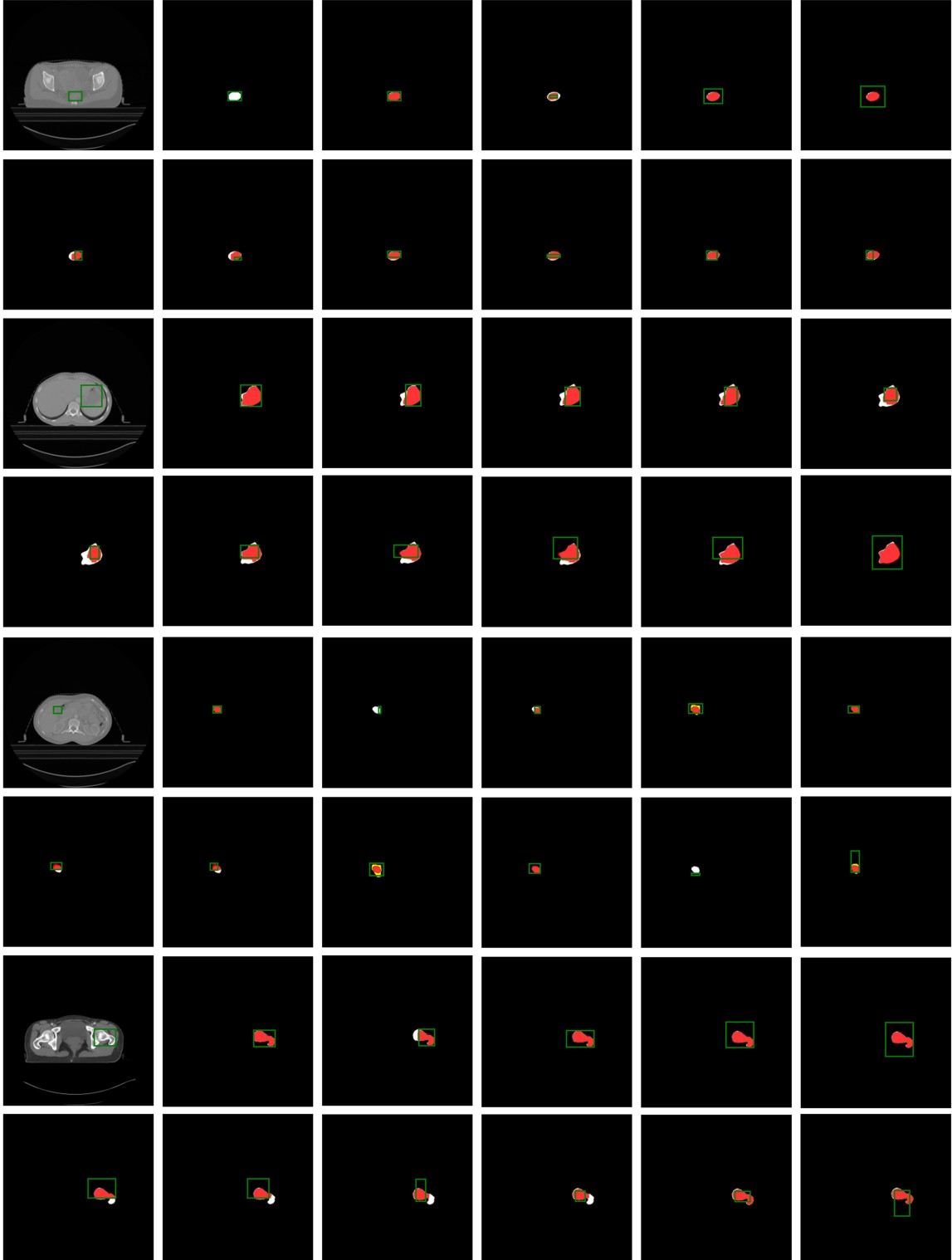

Figure 8: Noisy prompts. The green box refers to the box prompt. White refers to the ground-truth label. Red refers to the overlapping of the prediction and the GT label. Yellow refers to the incorrect prediction.

missed, and overly large box prompts may lead to incorrect predictions. However, slightly larger box prompts than the standard bounding boxes result in minimal impact. Therefore, we recommend making the box prompts slightly larger to avoid missing annotations as much as possible.

## D.2. Z-spacing

Our method works fine under common spacing settings (such as (1,1,3), (1,1,5)). Currently, the performance of our Slide SAM on consecutive slices with larger differences may be relatively poor. As shown in the Table 4, we resampled the volumes of the Word testset along the z-axis to change the z-spacing. We find that slightly increasing the z-spacing does not have much impact on performance, but performance notably decreases when the z-spacing becomes too large. We plan to improve this problem in future work, such as adding an adaptive module to Slide-SAM to try to robust the model identify variable slices that are further apart.

| Method | Z-spacing | Resampling ratio | Liver | Spleen | Pancreas | Gallbladder |
|---|---|---|---|---|---|---|
| Slide-SAM-H | $2.50\,mm \sim 3.00\,mm$ | 1.0 | 94.36 | 90.01 | 71.84 | 69.03 |
| | $2.75\,mm \sim 3.30\,mm$ | 1.1 | 94.99 | 89.10 | 71.18 | 69.31 |
| | $3.00\,mm \sim 3.60\,mm$ | 1.2 | 95.03 | 88.34 | 69.59 | 70.79 |
| | $3.75\,mm \sim 4.50\,mm$ | 1.5 | 94.51 | 89.17 | 68.36 | 68.10 |
| | $5.00\,mm \sim 6.00\,mm$ | 2.0 | 94.09 | 88.20 | 66.19 | - |
| | $6.25\,mm \sim 7.50\,mm$ | 2.5 | 93.29 | 89.15 | 60.66 | - |
| | $10.00\,mm \sim 12.00\,mm$ | 4.0 | 90.25 | 82.58 | 48.41 | - |
| | $15.00\,mm \sim 18.00\,mm$ | 6.0 | 85.84 | 75.20 | 42.38 | - |
| | $17.50\,mm \sim 21.00\,mm$ | 7.0 | 83.60 | 73.54 | - | - |
| | $18.00\,mm \sim 24.00\,mm$ | 8.0 | 81.71 | - | - | - |

Table 4: Evaluation on WORD testset (**Dice (%)**).

## D.3. Heterogeneous datasets

We conduct an additional testing on an in-house dataset sourced from the Guangdong Provincial People's Hospital in China. This dataset comprises 120 MRI volumes, partitioned into subsets: 100 volumes for training, 10 volumes for validation, and 10 volumes for testing purposes. The dataset comprises scans focusing on the lower abdomen, with the segmentation targets being the rectum and rectal tumors. One challenge of this dataset is its divergence from our pre-training data, which primarily consists of CT scans. Specifically, the CHAOS dataset is our sole MRI dataset, involving upper abdominal scans. Consequently, our model has not been exposed to lower abdominal MRI data previously. Furthermore, the spacing between slices in this dataset measures approximately (0.36, 0.36, 5), with a pronounced inter-slice gap. This structural peculiarity poses a significant challenge for our model's adaptation. To address these challenges, we employ a fine-tuning strategy using the available training data and subsequently evaluate the performance of our model. The results are shown in Table 5.

We find that when only one box prompt is used, the performance is not as good as nnUNet, but when we use 3 or more prompts, the performance can exceed nnUNet. Ad-

| Method | Prompt | Rectum | Tumor |
|---|---|---|---|
| nnUNet-2d | | 59.62 | 47.56 |
| nnUNet-3d | | 68.48 | 61.68 |
| Slide-SAM-finetune | 1 box | 65.10 | 52.89 |
| | 3 boxes | 75.92 | 61.85 |
| | 5 boxes | 81.11 | 64.48 |
| | 10 boxes | **83.96** | **65.12** |

Table 5: Evaluation on Rectal in-house dataset. (Dice (%))

ditionally, the segmentation accuracy can be further improved as the number of prompts increases.

**D.4. Pseudo labels:**

To prove the efficacy of pseudo-labels, we conducte a comparative analysis by assessing model performance with and without their utilization. As shown in Table 6, we employ all slices from the 3D volumes of AbdomenCT-1K as the training images and finetune SAM with the incorporation of the LoRA module. Subsequently, validation is carried out on the test set of AbdomenCT-1K. Our findings indicate that, when using only the original data and their associated labels, the finetuned model's performance is even inferior to that of the original SAM. However, a notable enhancement in performance is observed when we incorporate the additional pseudo-labels, thereby affirming the constructive impact of the pseudo-labels used in our model training.

| mIoU | Dataset for ft. | Point | Box |
|---|---|---|---|
| SAM | | 56.16 | 72.36 |
| SAM (ft.) | Abd-1K | 45.68 | 56.06 |
| SAM (ft.) | + pseudo masks | 66.82 | 74.87 |

Table 6: Comparison between the utilization of generated pseudo-labels and their absence on AbdomenCT-1K testset.

