# OpenReview forum: "Slide-SAM: Medical SAM Meets Sliding Window"
_MIDL.io/2024/Conference — MIDL 2024 Oral_

### Official Review · Reviewer_k6BF · 2024-02-26

**Confidence:** 4
**Preliminary Rating:** 5
**Recommendation:** Oral

**Summary:**

The paper introduces Slide-SAM, an adaptation of the Segment Anything Model (SAM) for 3D medical image segmentation. Unlike SAM, which struggles with the contextual relationships in 3D medical images, Slide-SAM uses a stack of three slices to predict segmentation masks, improving efficiency and accuracy. It demonstrates effectiveness across various datasets.

**Strengths:**

Slide-SAM introduces a novel method for 3D medical image segmentation, leveraging a sliding window technique with minimal prompting for efficient segmentation. This approach significantly improves upon the limitations of existing 2D models by adapting them for 3D contexts, demonstrating versatility across multiple medical datasets. Its innovative use of adjacent slice information to guide segmentation offers a practical solution to the challenge of understanding complex 3D medical images, showcasing the model's potential for broad applicability in the medical imaging field.

**Weaknesses:**

While Slide-SAM represents a significant advancement, its reliance on the continuity between adjacent slices may not be universally applicable across all types of medical images, where variability between slices can be substantial. Additionally, the model's performance in highly heterogeneous datasets remains to be fully explored. The computational efficiency of the sliding window approach, especially for large volumes, and the potential for increased processing time or resource demands, are concerns that could limit its practical deployment in resource-constrained settings.

**Detailed Comments:**

Not comments

**Justification Of The Preliminary Rating:**

The innovative approach of Slide-SAM addresses a significant gap in medical image segmentation, offering a promising solution to the challenges of 3D image analysis. Despite potential limitations in adaptability and computational demand, its successful application across diverse datasets justifies a positive preliminary assessment.

**Questions To Address In The Rebuttal:**

No comments

**Special Issue:**

No

---

> ### Author Response · Authors · 2024-03-12
>
> Thanks for your advice, which could impove Slide-SAM robustness and make it able to face the challenges of variable clinical applications.
>
> We have added all the following discussions in appendix.
>
> ## Slice variability:
> Our method works fine under common spacing settings (such as (1,1,3), (1,1,5)). Currently, the performance of our Slide SAM on consecutive slices with larger differences may be relatively poor. As shown in the table below, we resampled the volumes of the Word testset along the z-axis to change the z-spacing. We find that slightly increasing the z-spacing does not have much impact on performance, but performance notably decreases when the z-spacing becomes too large. We plan to improve this problem in future work, such as adding an adaptive module to Slide-SAM to try to robust the model identify variable slices that are further apart.
>
> |  Dataset     | z-spacing       | liver | spleen | pancreas | gallbladder |
> |--------------|-----------------|-------|--------|----------|-------------|
> | WORD testset | 2.75mm   | 94.36	 | 90.01	 | 71.84	 | 69.03       |
> |              | 3.00mm   | 94.99 | 89.1   | 71.18    | 69.31       |
> |              | 3.30mm   | 95.03 | 88.34  | 69.59    | 70.79       |
> |              | 4.12mm   | 94.51 | 89.17  | 68.36    | 68.1        |
> |              | 5.50mm   | 94.09 | 88.2   | 66.19    | -           |
> |              | 6.87mm   | 93.29 | 89.15  | 60.66    | -           |
> |              | 11.00mm | 90.25 | 82.58  | 48.41    | -           |
> |              | 16.50mm | 85.84 | 75.2   | 42.38    | -           |
> |              | 19.25mm | 83.60 | 73.54   | -   | -           |
> |              | 22.00mm | 81.71 | -   | -    | -           |
>
> ## Heterogeneous datasets:
>
> We conduct an additional testing on an in-house dataset sourced from the Guangdong Provincial People’s Hospital in China. This dataset comprises 120 MRI volumes, partitioned into subsets: 100 volumes for training, 10 volumes for validation, and 10 volumes for testing purposes. The dataset comprises scans focusing on the lower abdomen, with the segmentation targets being the rectum and rectal tumors.
> One challenge of this dataset is its divergence from our pre-training data, which primarily consists of CT scans. Specifically, the CHAOS dataset is our sole MRI dataset, involving upper abdominal scans.
> Consequently, our model has not been exposed to lower abdominal MRI data previously. Furthermore, the spacing between slices in this dataset measures approximately (0.36, 0.36, 5), with a pronounced inter-slice gap. This structural peculiarity poses a significant challenge for our model's adaptation.
> To address these challenges, we employ a fine-tuning strategy using the available training data and subsequently evaluate the performance of our model. The results are shown in Table below.
>
> We find that when only one box prompt is used, the performance is not as good as nnUNet, but when we use 3 or more prompts, the performance can exceed nnUNet. Additionally, the segmentation accuracy can be further improved as the number of prompts increases.
>
> | Method             | Prompt   | Rectum | Tumor |
> |--------------------|----------|--------|-------|
> | nnUNet-2d          |          | 59.62  | 47.56 |
> | nnUNet-3d          |          | 68.48  | 61.68 |
> | Slide-SAM-finetune | 1 box    | 65.10  | 52.89 |
> |                    | 3 boxes  | 75.92  | 61.85 |
> |                    | 5 boxes  | 81.11  | 64.48 |
> |                    | 10 boxes | 83.96  | 65.12 |
>
>
>
> ## Computational efficiency:
>
> We adopt a parallel strategy during inference, that is, divide all adjacent slices into multiple batches, and increase the batch size as large as possible to ensure that as much GPU memory is utilized as possible. Inference is performed on Nvidia Titan RTX 24G and the batchsize is set to 4. For Slide-SAM, the running speed is 3.25 sec/volume, and the GPU memory usage is 14G. For Slide-SAM-H, the running speed is about 10 sec/volume, and the GPU memory usage is 17G. In future, we will also try more mobile-friendly technologies to light-weight and accelerate model inference, such as using EfficientViT as backbone or SAMI pretrain strategy in EfficientSAM, using model compression and distillation technology or other operator acceleration technologies, etc.

---

> > ### Comment · Reviewer_k6BF · 2024-03-18
> >
> > Thanks for addressing those concerns. I would accept it now.

---

### Official Review · Reviewer_CJn3 · 2024-02-28

**Confidence:** 5
**Preliminary Rating:** 5
**Recommendation:** Oral
**Final Rating:** 5

**Summary:**

The paper presents Slide-SAM to address the challenge of segmenting 3D medical images with minimal prompts. Slide-SAM utilizes a sliding window of three adjacent slices, beginning with a prompt on the central slice and generating successive prompts through prediction masks for the entire volume, leveraging pre-trained SAM weights and a hybrid loss function to efficiently train on both 2D and 3D data. Extensive experiments on various datasets demonstrate that Slide-SAM achieves superior performance over existing methods with fewer required prompts.

**Strengths:**

* Slide-SAM’s approach of processing images in a stacked sequence allows it to efficiently infer the whole volume from minimal starting information.

*  The system is adept at leveraging a wide range of dataset types, training with a mix of complete 3D labels and 2D generated labels.

* The proposed model outperforms previous segmentation methods by achieving higher accuracy with a reduced number of prompts.

* The framework is publicly accessible.

**Weaknesses:**

The paper is well-written. The paper may have tested only certain types and levels of noise. In real-world clinical settings, noise and inaccuracies in prompts can be more unpredictable and diverse. Hence, if the range of noise conditions tested doesn't cover the full spectrum encountered in clinical practice, sensitivity to the initial input could remain a concern (but may a bit of out-of the scope in this manuscript)

**Detailed Comments:**

* The robustness of predictions against noisy prompts might vary across different types of anatomical structures, especially those with less distinct boundaries or more complex shapes.

* The practicality of integrating such an advanced tool into routine medical practice may raise concerns. These include the time and expertise clinicians need to provide initial prompts and to merge the tool into existing diagnostic workflows. The authors might consider adding a discussion to mitigate the learning curve for researchers and clinicians.

* Please add more description in Figure-1's caption.

**Justification Of Final Rating:**

The paper demonstrates the model's superior performance quantitatively through extensive benchmarking and experiments. The authors have effectively addressed my comments, and with the paper now in excellent condition.

**Justification Of The Preliminary Rating:**

The paper demonstrates the model's superior performance quantitatively through extensive benchmarking and experiments. The results show clear enhancements in annotation efficiency, which is imperative for making such technologies viable in clinical settings.

**Questions To Address In The Rebuttal:**

Please check the weakness and detailed comments section.

**Special Issue:**

Yes

---

> ### Author Response · Authors · 2024-03-12
>
> Thank you for your valuable reviews, which greatly contributes to refining our work and its practical application in real clinical settings.
>
> (1) We have added the discussion about more noisy prompts in Appendix. As we are unable to share images, we provide textual descriptions instead. Through our testing, we have discovered that certain types of noise adversely affect prediction accuracy when the box prompt is completely contained within the ground truth bounding box. Optimal performance is achieved when the box prompt slightly exceeds the bounding box. Particularly challenging organs or lesions are more susceptible to noise interference. To address these issues comprehensively, we plan to conduct further testing and optimization in future iterations, ensuring our model's efficacy in diverse real-world scenarios. Our work has been open sourced in github (https://github.com/Curli-quan/Slide-SAM) and the model weights have been released. We sincerely hope that our contribution benefits the medical imaging community, and you are welcome to use our open source project and discover more problems.
>
> (2) We have added a discussion in Appendix about usage for clinicians. We are exploring the integration of Slide-SAM into existing labeling tools, such as Slicer, as a plug-in. Furthermore, we will provide comprehensive guidelines to assist researchers and clinicians in effectively utilizing our tool.
>
> (3) We have added more description in Figure-1's caption in the revised manuscript.

---

> > ### Comment · Reviewer_k6BF · 2024-03-18
> >
> > Thanks for addressing those concerns. I would accept it now.

---

### Official Review · Reviewer_pAdm · 2024-02-28

**Confidence:** 4
**Preliminary Rating:** 4
**Recommendation:** Oral
**Final Rating:** 5

**Summary:**

SAM struggles to learn contextual relationships between slices when applied to 3D medical images. Moreover, applying 2D SAM to 3D images requires prompting the entire volume. In this paper, the authors propose Slide-SAM, which only requires a prompt from the central slice to simultaneously infer multiple adjacent slices, and the resulting predictions can be used to generate prompts for the next group of adjacent slices, which could better adapt SAM to 3D medical image segmentation tasks.

**Strengths:**

The motivation is clear and interesting. The proposed method is evaluated on relatively large-scale datasets with comparison to other recent state-of-the-art methods to demonstrate the effectiveness and generalizability.

**Weaknesses:**

Several minor suggestions for the article.

Table 1. It would be better to add the average performance of our target organs. Similar for Table 3.
Besides, comparison with SAM-Med3D, another adaption of SAM for 3D medical images which could also enable segmentation with single prompt, should be included and compared.
Minor typos Sec 3.1 training data should be “TotalSegmentator”

**Detailed Comments:**

Table 1. It would be better to add the average performance of our target organs. Similar for Table 3.
Besides, comparison with SAM-Med3D, another adaption of SAM for 3D medical images which could also enable segmentation with single prompt, should be included and compared.
Minor typos Sec 3.1 training data should be “TotalSegmentator”

**Justification Of Final Rating:**

Thanks for addressing the comments. Great work!  ( Another minor question in the revised version of manuscript. Please ensure that the significant figures in Table 1 are consistent, and add the prompt settings in line SAM-Med3d

**Justification Of The Preliminary Rating:**

The manuscript introduces novel design components to further adapt SAM for 3d medical image segmentation tasks. The designs are sound with comprehensive validation and offer good impact. Please address the issues mentioned above.

**Questions To Address In The Rebuttal:**

See above.

**Special Issue:**

No

---

> ### Author Response · Authors · 2024-03-12
>
> Thanks for your reviews, which greatly contributes to enhancing the completeness of our work and experiments. We have evaluated SAM-Med3D using the official checkpoint and summarized the results in the table below. It is worth noting that the SAMMed-3D method only supports point prompts, whereas our approach can utilize either point prompts or box prompts. The ability to use box prompts represents an advantage of our method over SAMMed-3D. Therefore, in the performance comparison, we contrasted our performance using box prompts with SAMMed-3D's performance using point prompts. It is evident that our approach yields better results when compared to SAMMed-3D, further affirming the effectiveness and potential of Slide-SAM.
>
> Additionally, we have added the average performance and fixed all typos in the revised manuscript.
>
> | WORD testset |          |       |           |           |           |         |             |           |           |
> |--------------|----------|-------|-----------|-----------|-----------|---------|-------------|-----------|-----------|
> | Method       | Prompt   | Liver | Spleen    | Kidney(L) | Kidney(R) | Stomach | Gallbladder | Esophagus | Pancrease |
> | Ours         | 1 box    | 94.36 | 90.01     | 93.04     | 91.39     | 76.46   | 69.03       | 60.75     | 71.84     |
> | SAMMed-3D    | 1 point  | 90.38 | 82.83     | 84.1      | 87.24     | 55.28   | 60.56       | 28.02     | 43.49     |
> | Method       | Duodenum | Colon | Intestine | Adrenal   | Rectum    | Bladder | Femur(L)    | Femur(R)  | Average   |
> | Ours         | 50       | 50.15 | 56.78     | 33.98     | 77.86     | 84.11   | 93.47       | 93.72     | 74.18     |
> | SAMMed-3D    | 31.41    | 24.45 | 30.47     | 0.86      | 60.38     | 86.84   | 83.89       | 73.65     | 57.74     |
>
> |  CHAOS testset           |         ||        |        |        |         |
> |-----------|---------|---------------|--------|--------|--------|---------|
> | Method    | Prompt  | Liver         | Kid(R) | Kid(L) | Spleen | Average |
> | Ours      | 1 box   | 88.42         | 91.89  | 90.98  | 90.76  | 90.51   |
> | SAMMed-3D | 1 point | 82.8          | 80.25  | 78.49  | 79.37  | 80.22   |
>
> | BTCV testset |         |       |         |        |        |         |
> |--------------|---------|-------|---------|--------|--------|---------|
> | Method       | Prompt  | Liver | Stomach | Kid(L) | Spleen | Average |
> | Ours         | 1 box   | 95.44 | 74.31   | 92.42  | 91.16  | 88.33   |
> | SAMMed-3D    | 1 point | 64.31 | 54.73   | 85.98  | 75.38  | 70.10   |

---

> > ### Comment · Reviewer_k6BF · 2024-03-18
> >
> > Thanks for addressing the comments. I would accept this paper now.

---

### Meta-Review · Area_Chair_vEdK · 2024-03-30

**Recommendation:** Accept (Oral)
**Confidence:** 5

**Metareview:**

Reviewers unanimously agree to strongly accept the submission.

---

### Decision · Program_Chairs · 2024-04-05

Accept (Oral)